# Changes in Gastric Pathology after *H. pylori* Treatment in Community-Driven Research Aimed at Gastric Cancer Prevention

**DOI:** 10.3390/cancers15153950

**Published:** 2023-08-03

**Authors:** Ting Wang, Safwat Girgis, Hsiu-Ju Chang, Ali Assi, Katharine Fagan-Garcia, Taylor Cromarty, Rachel Munday, Karen J. Goodman, Sander Veldhuyzen van Zanten

**Affiliations:** 1Department of Medicine, Faculty of Medicine & Dentistry, University of Alberta, Edmonton, AB T6G 2X8, Canada (the CAN*Help* Working Group); 2Department of Lab Medicine & Pathology, Faculty of Medicine & Dentistry, University of Alberta, Edmonton, AB T6G 2X8, Canada; 3Susie Husky Health Centre, Aklavik, NT X0E 0A0, Canada

**Keywords:** *Helicobacter pylori*, Arctic, Canada, indigenous health, gastritis, active gastritis, chronic gastritis, atrophic gastritis, intestinal metaplasia, gastric cancer

## Abstract

**Simple Summary:**

Community-driven projects have characterized health impacts of *Helicobacter pylori* (*Hp*) infection in Indigenous communities in the Northwest Territories (NT) and Yukon (YT), Canada. These projects address concerns expressed by community leaders about the frequent diagnosis of *Hp* infection and its relation to gastric cancer deaths, perceived to occur with alarming frequency in this region. Previous project results showed a high prevalence of stomach pathologies associated with increased cancer risk among *Hp*-positive participants at baseline. This follow-up study investigated changes in precancerous stomach pathologies after treatment to eliminate *Hp* infection among 69 project participants with follow-up data. Most participants who had successful treatment to eliminate *Hp* at baseline remained infection-free at follow-up and the prevalence of precancerous stomach pathologies was lower at follow-up than baseline. The more frequent improvements in precancerous stomach pathologies observed in participants who were *Hp*-negative at follow-up relative to those who were *Hp*-positive at follow-up suggests that treatment to eliminate *Hp* infection has the potential to reduce stomach cancer risk in participating communities.

**Abstract:**

Community-driven projects have characterized *Helicobacter pylori* (*Hp*) infection in Indigenous communities in the Northwest Territories (NT) and Yukon (YT), Canada. These projects address concerns about the frequent diagnosis of *Hp* infection among community members and its relation to gastric cancer deaths, perceived to occur with alarming frequency in this region. Projects included breath-test screening for *Hp* infection, gastroscopy with gastric biopsies, and treatment to eliminate *Hp* infection. Previous project results showed a high prevalence of stomach pathologies associated with increased cancer risk among *Hp*-positive participants at baseline. This analysis describes changes in precancerous gastric pathologies in project participants who had gastroscopy before baseline treatment during 2008–2013 and again in 2017. Throughout the study period, the same pathologist graded *Hp* density, active gastritis, chronic gastritis, atrophic gastritis, and intestinal metaplasia using the updated Sydney System. Of 310 participants from three communities with baseline pathology data, 69 had follow-up pathology data. Relative to baseline, the prevalence of *Hp* infection and precancerous gastric pathology was substantially lower at follow-up; most participants who were *Hp*-positive at baseline and *Hp*-negative at follow-up had reduced severity of active, chronic, and/or atrophic gastritis at follow-up. Though follow-up numbers are small, these results yield evidence that successful *Hp* treatment has the potential to reduce the risk of gastric cancer in Arctic Indigenous communities.

## 1. Introduction

*Helicobacter pylori* (*Hp*), responsible for a large burden of disease across the globe, is a bacterium that inhabits the gastric epithelium [1], causing chronic inflammation in the stomach, which can lead to dyspepsia, peptic ulcer disease and gastric cancer [2]. Most *Hp* infections are acquired in childhood and often persist long term, going undetected due to the absence of symptoms in most cases, or non-specific dyspeptic symptoms in a small fraction of cases [2,3]. While questions about modes of transmission remain, the body of evidence points to direct person-to-person transmission through contact with digestive fluids as the usual pathway [3]. A 2015 analysis estimated that *Hp* infection causes 89% of non-cardia gastric carcinomas, estimated at 780,000 new cases diagnosed in 2008, constituting around 6% of all new cancers diagnosed worldwide [4].

The current understanding of gastric carcinogenesis posits a cascade of progressively degenerative changes arising from chronic gastritis, as described by Correa and Piazuelo [5]. The sequence of gastric carcinoma precursors that follow chronic gastritis includes atrophic gastritis, intestinal metaplasia, and dysplasia, with progression influenced by bacterial virulence interacting with host and environmental factors such as diet and smoking [6].

The prevalence of *Hp* has been declining globally over the past century, especially in areas with modern sanitation infrastructure [7,8,9,10,11,12,13,14,15]. However, comparable declines in prevalence are not evident in less-developed regions [11,16,17]. Epidemiologic studies in diverse countries show that educational attainment and other indicators of socioeconomic status typically have inverse associations with *Hp* prevalence [18,19,20,21]. Indigenous communities in the circumpolar region of Alaska, Greenland, Russia, and Canada have been shown to have a disproportionately high occurrence of *Hp* infection and gastric cancer [22]. In Canada’s Northwest Territories (NT) during 1997–2015, estimated incidence rates of non-cardia gastric cancer in Indigenous men and Indigenous women were 2.7 and 3.0 times, respectively, the rates of this cancer in their counterparts across Canada [23]. These estimates align with concerns expressed by Indigenous community leaders in northern Canada about the frequent diagnosis of *Hp* infection and its relation to gastric cancer deaths, which they perceived to occur with alarming frequency in this region. Beginning in the early 2000s, leaders of Indigenous communities in the NT and neighboring Yukon (YT) have advocated for research aimed at reducing the cancer risk from *Hp* infection.

Since 2008, community-driven projects have characterized *Hp* infection and its health impacts in Indigenous NT and YT communities to address community concerns [24,25,26,27]. Conducted by the Canadian North *Helicobacter pylori* (CAN*Help*) Working Group, a collaboration of community leaders and healthcare providers with University of Alberta researchers, each project has been led by a planning committee of community stakeholders. Community projects have included screening for and treatment of *Hp* infection, as well as gastroscopy to characterize gastric health before and after treatment. The objective of this analysis was to describe changes in gastric pathology after treatment to eliminate *Hp* infection in community-driven research aimed at gastric cancer prevention.

## 2. Materials and Methods

### 2.1. Participating Communities

In 2017, we offered follow-up gastroscopy to residents of 3 Canadian Arctic communities who had a baseline gastroscopy in a CAN*Help* community project [28]. Baseline gastroscopies occurred in 2008 in Aklavik, NT, 2012 in Old Crow, YT, and 2013 in Fort McPherson, NT (Figure 1). The initial project in Aklavik was the best resourced with the largest number of participants [24]. Its success led leaders of neighboring communities to invite the research team to launch CAN*Help* projects in their communities. Using the Aklavik project as a model, each community project planning committee decided how to adapt it to their setting; all planning committees chose to prioritize comparability to other CAN*Help* projects. Aklavik’s population was estimated at approximately 600 around 2008, predominantly Inuvialuit (western Canadian Inuit) or Gwich’in (Athabaskan) First Nations. It is situated 113 km south of the Arctic Coast, accessible by air, as well as ice road in the winter or the Mackenzie River in the summer [29,30,31,32]. The second project, in Old Crow, the northernmost YT community (216 km west of Aklavik), achieved the highest participation proportion of its population, estimated at roughly 250 around 2010, nearly all belonging to the Vuntut Gwitchin (Athabaskan) First Nation. Old Crow is situated on the Porcupine River, north of the Arctic Circle, accessible only by aircraft or canoe [33]. Fort McPherson’s population was estimated at roughly 800 around 2012, predominantly Gwich’in (Athabaskan) First Nations. It is situated on the Peel River, accessible by air, river, and highway, 91 km south of Aklavik [29,31,34].

### 2.2. Summary of Research Sought by Communities

Adhering to principles of community-driven research [25], project planning committees guided the research focus, implementation logistics, and interpretation of results. They reviewed study protocols and questionnaires to ensure cultural appropriateness. They also reviewed reports of research results before they were made public to provide input on interpretation. All residents of the 3 communities were eligible to participate, with “resident” defined as being present in the community at the time gastroscopy was offered.

### 2.3. Study Population and Design

Approximately 50% of the combined population of participating communities enrolled in the local CAN*Help* project (~64% in Aklavik, ~85% in Old Crow, ~30% in Fort McPherson) [27]. Baseline project activities included screening for *Hp* infection using the 13-C urea breath test (UBT) [35]. The fraction of the census population with data on *Hp* status was 59% in Aklavik, 81% in Old Crow, and 28% in Fort McPherson [27]. We collected participant data using interviewer-administered questionnaires and health center chart reviews. We invited all participants ≥15 years of age to have gastroscopy with gastric biopsies regardless of *Hp* status; youth under 15 years of age were included at the request of parents.

Before our first community project launched, local clinicians had observed an unacceptably low effectiveness of the conventional triple therapy (clarithromycin and amoxicillin with a proton-pump inhibitor) used widely across Canada to cure *Hp* infection. Thus, a key aim of the community projects was to identify a more effective treatment regimen. We invited *Hp*-positive participants to enroll in treatment trials, described in detail previously [36]. In the first community project, we compared the standard therapy (standard doses of amoxicillin, clarithromycin and rabeprazole for 10 days) against a sequential therapy (days 1–5, standard doses of amoxicillin and rabeprazole, days 6–10 standard doses of clarithromycin, metronidazole and rabeprazole) or a bismuth-based quadruple therapy (standard doses of tetracycline, metronidazole, bismuth and rabeprazole for 10 days) [36]. We discontinued the standard therapy due to poor effectiveness in the first community treatment trial; in subsequent community trials, we compared the sequential and bismuth-based quadruple therapies [36]. We assessed the treatment outcome by UBT performed at least 8 weeks after the treatment regimen was completed; we defined treatment success as a negative result and treatment failure as a positive or uncertain result.

### 2.4. Diagnostic Methods

Details of diagnostic methods used in CAN*Help* projects have been described previously [27]. In brief, gastroenterologists conducted unsedated gastroscopies in temporary endoscopy units set up in each community’s health center, collecting 5 gastric biopsies from each participant for histopathology (2 each from antrum and corpus, 1 from incisura) and 2 for culture. We transported biopsies fixed in formalin to the University of Alberta. One pathologist (SG) evaluated all biopsy samples based on the updated Sydney protocol, grading *Hp* density in ordinal categories of none, mild, moderate or marked, and the severity of active gastritis, chronic gastritis, atrophic gastritis, and intestinal metaplasia in ordinal categories of none, mild, moderate, and severe [37]. We used E-TESTs^®^ to detect resistance to 7 antibiotics (metronidazole, clarithromycin, tetracycline, amoxicillin, nitrofurantoin, refampicin, and ciprofloxacin) in *Hp* isolated from culture of gastric biopsies.

Due to resource constraints, we were not able to offer follow-up gastroscopy at uniform follow-up intervals across communities. Instead, we invited participants who completed a gastroscopy during 2008–2013 to have a follow-up gastroscopy in 2017. The biopsy collection protocol remained the same, except an additional biopsy was taken from the incisura, for a total of 6 biopsies taken for histopathology. The same pathologist evaluated the biopsies in the manner used for the baseline assessment.

### 2.5. Statistical Analysis

We estimated the prevalence of *Hp* infection (detected by histopathology) and histopathologic abnormalities (active gastritis, chronic gastritis, atrophic gastritis, and intestinal metaplasia) at baseline and follow-up. We present the proportion in each severity category, with participants classified based on the highest severity observed in multiple biopsies. To examine the change in disease severity, we estimated the proportion of participants with increased, decreased or no change in the severity of each histopathologic classification, comparing follow-up to baseline. We also estimated the probability of changes in gastric pathology including improvement in the severity of chronic gastritis, active gastritis, and atrophic gastritis among participants who had these conditions detected at baseline, as well as progression in the severity (including new onsets) of atrophic gastritis and/or intestinal metaplasia among all participants (none of whom had severe atrophic gastritis or severe intestinal metaplasia at baseline). We present all proportions with 95% confidence intervals (CI) to indicate the precision of the estimate. We estimated the risk difference and 95% confidence interval as a measure of association based on the difference in the probability of changes in gastric pathology between categories of selected participant characteristics. We examined the following participant characteristics: age; sex; ethnicity (Inuvialuit, Gwich’in, other Indigenous, non-Indigenous); community of residence; time in years between baseline and follow-up gastroscopy; baseline *Hp* density; baseline antimicrobial resistance; baseline pathology severity; completion of treatment to eliminate *Hp* at baseline; post-treatment *Hp* status; *Hp* density at follow-up; antimicrobial resistance at follow-up; and improvement of *Hp* density from baseline to follow-up. To control for confounding, we estimated risk differences in multivariable binomial regression models. We used the purposeful selection method to select variables for regression models [38]. We used the statistical software package STATA 15 for analysis. This research was approved by the University of Alberta Health Research Ethics Board—Biomedical Panel (study ID Pro00007868).

## 3. Results

### 3.1. Participant Characteristics

Combining the three communities, 310 participants (~19% of the population of the three communities combined) had baseline gastroscopy with pathological assessment of gastric biopsies; 69 (22%) of these 310 participants had follow-up gastroscopy. Table 1 compares characteristics of the baseline and follow-up study populations. Most participants were in the 30–59-year age range at baseline (55%) and follow-up (59%), though a smaller proportion were under 30 and a larger proportion over 59 in the follow-up group, as would be expected with the passage of time. More participants were female at baseline (55%) and follow-up (64%). The proportion of Indigenous participants was 93% at baseline and 96% at follow-up, with slightly greater participation in follow-up of Gwich’in participants relative to other ethnicities.

### 3.2. Follow-Up Time Period

The average follow-up period for participants was 7.2 years, with a median and mode of 9.1 years, minimum of 4.2 years and maximum of 9.3 years.

### 3.3. Hp Prevalence and Antimicrobial Resistance at Baseline and Follow-Up

Table 2 shows the prevalence of *Hp*, classified by histopathologic assessment of gastric biopsies, at baseline (72% [223/310]) and follow-up (26% [18/69]). Of note, the prevalence of *Hp* infection at baseline among the 69 follow-up participants was 75%, similar to the baseline prevalence among all 310 participants. Antimicrobial resistance status was available for 207 participants classified as *Hp*-positive by histology at baseline. The frequency of resistance was 34% (95% CI: 28, 41%) for metronidazole and 16% (95% CI: 12, 22%) for clarithromycin; 43% (95% CI: 36, 49%) of participants were resistant to one or more antibiotics and 12% (95% CI: 8, 17%) to two or more antibiotics. Among the 69 follow-up participants, 48 had antimicrobial resistance status for *Hp* infection at baseline. The follow-up group had similar baseline resistance frequencies: 40% (95% CI: 27, 54%) for metronidizole, 15% for clarithromycin (95% CI: 7, 28%), 44% (95% CI: 30, 58%) for one or more antibiotics and 15% (95% CI: 7, 28%) for two or more antibiotics. Antimicrobial resistance status at follow-up was available for 15 of the 18 participants who were *Hp*-positive at follow-up. One of the fifteen (7%, 95% CI: 1, 40%) was resistant to metronidazole, four (27%, 95% CI: 9, 56%) were resistant to clarithromycin and none were resistant to more than one antibiotic.

### 3.4. Prevalence of Abnormal Gastric Pathology at Baseline and Follow-Up

Table 2 also shows the prevalence of active gastritis, chronic gastritis, atrophic gastritis and intestinal metaplasia, by severity at baseline and follow-up. No cases of gastric dysplasia or carcinoma were detected at baseline or follow-up. Active gastritis prevalence decreased from 70% at baseline to 26% at follow-up. The prevalence of all active gastritis severity levels was lower at follow-up than at baseline: mild active gastritis prevalence decreased from 33% to 17%; moderate active gastritis prevalence decreased from 26% to 9%; severe active gastritis prevalence decreased from 11% to 0. Chronic gastritis prevalence decreased from 75% at baseline to 38% at follow-up. Moderate and severe chronic gastritis prevalence were lower at follow-up than baseline, while mild chronic gastritis prevalence increased: moderate chronic gastritis prevalence decreased from 32% to 16%; severe chronic gastritis prevalence decreased from 34% to 4%; and mild chronic gastritis prevalence increased from 9% to 17%. Atrophic gastritis prevalence decreased from 31% at baseline to 14% at follow-up. All atrophic gastritis severity levels decreased in prevalence: mild decreased from 22% to 12%, moderate decreased from 7% to 3% and severe decreased from 2% to 0. In contrast, intestinal metaplasia prevalence increased slightly, from 14% at baseline to 19% at follow-up: mild intestinal metaplasia increased in prevalence from 9% to 13%, moderate intestinal metaplasia prevalence increased from 4% to 6% and severe intestinal metaplasia prevalence decreased from 1% to 0.

### 3.5. Within-Individual Change in Abnormal Gastric Pathology Severity from Baseline to Follow-Up

Table 3 shows within-individual change in severity of histopathology from baseline to follow-up, stratified by *Hp* status at baseline and follow-up. Of the 69 participants, 52 were classified as positive at baseline, 47 (90%) of whom received treatment to eliminate *Hp* infection following the baseline gastroscopy. Of these 47, baseline treatment was successful in 34 (72%) and failed in 13 (28%). At follow-up, 51 (74%) participants were classified as *Hp*-negative and 18 (26%) as *Hp*-positive. Participants who were *Hp*-negative at follow-up included 16 of 17 (94%) participants who were *Hp*-negative at baseline, 27 of 34 (79%) with successful treatment at baseline, 6 of 13 (46%) with failed treatment at baseline and 2 of 5 (40%) who were *Hp*-positive at baseline and did not receive treatment. Participants who were *Hp*-positive at follow-up included 1 of 17 (6%) participants who were *Hp*-negative at baseline, 7 of 34 (21%) with successful treatment at baseline, 7 of 13 (54%) with failed treatment at baseline, and 3 of 5 (60%) who were *Hp*-positive at baseline and did not receive treatment. To avoid overemphasizing imprecise distinctions due to small group sizes, we stratified changes in gastric pathology only by *Hp* status at baseline and follow-up, without further stratifying by treatment status or treatment response. This data presentation makes sense because some participants with negative post-treatment breath tests may have had suppressed *Hp* infections that were not fully eliminated; in cases of reinfection, the time of reinfection onset is unknown; some participants could have received treatment elsewhere without our knowledge. Table A1 and Table A2 in Appendix A include details for subgroups based on treatment status and treatment response.

Among the 17 participants who were *Hp*-negative at baseline and follow-up, none had active gastritis detected at baseline or follow-up and six (35%) had chronic gastritis detected at baseline or follow-up; among these six, chronic gastritis severity at follow-up was higher in three, unchanged in one and lower in two. Two of the seventeen (12%) participants had atrophic gastritis and intestinal metaplasia that was higher in severity at follow-up. Among the 34 participants who were *Hp*-positive at baseline and *Hp*-negative at follow-up, all 34 (100%) had chronic gastritis at baseline and 34 of 34 (100%) had lower chronic gastritis severity at follow-up; 33 (97%) had active gastritis at baseline and 33 of 33 (100%) had lower active gastritis severity at follow-up. Of these 34 participants, 14 (41%) had atrophic gastritis at baseline and 13 of 14 (93%) had lower atrophic gastritis severity at follow-up. Four of the thirty-four (12%) participants had intestinal metaplasia at baseline; intestinal metaplasia severity at follow-up was higher in two, unchanged in one and lower in one. Among the 18 participants who were *Hp*-positive at follow-up, all 18 (100%) had chronic gastritis and active gastritis at baseline; chronic gastritis severity at follow-up was lower in 7 (39%) and unchanged or higher in 11 (61%), while active gastritis severity at follow-up was lower in 5 (28%) and unchanged or higher in 13 (72%). In this group, 11 (61%) participants had atrophic gastritis at baseline; atrophic gastritis severity at follow-up was lower in 6 of 11 (55%) and unchanged or higher in 5 (45%). Seven participants with no atrophic gastritis observed at baseline had atrophic gastritis at follow-up. Nine of the eighteen (50%) had intestinal metaplasia at baseline; intestinal metaplasia severity at follow-up was lower in one of nine (11%), and unchanged or higher in eight of nine (89%). Eight participants with no intestinal metaplasia observed at baseline had intestinal metaplasia at follow-up.

### 3.6. Probability of Changes in Gastric Pathology Severity by Participant Characteristics

Table 4 shows the estimated probability of change in gastric pathology by selected participant characteristics. The small number of participants with follow-up data permitted the detection of non-random differences in the probability of changes in severity only when differences were quite large. Among 54 participants with chronic gastritis detected at baseline, 43 (80%) had a lower chronic gastritis severity classification (including none) at follow-up. Large differences in the probability of improved chronic gastritis were not observed by age, sex, ethnicity, community, baseline *Hp* resistance status, or time between baseline and follow-up gastroscopies. The probability of improved chronic gastritis severity increased with increased baseline *Hp* density as well as increased baseline chronic gastritis severity, perhaps because higher baseline severity corresponds to more opportunity for improvement. The frequency of improved chronic gastritis severity was relatively large among participants who completed treatment to eliminate *Hp* before follow-up, tested negative for *Hp* before follow-up, showed improvement in *Hp* density from baseline to follow-up and/or had no *Hp* observed in follow-up gastric biopsies. Among 50 participants with active gastritis detected at baseline, 36 (72%) had a lower active gastritis severity classification (including none) at follow-up. The frequency of improved active gastritis severity showed a similar pattern of association with participant characteristics as that observed for chronic gastritis, except it appeared more strongly associated with *Hp* density at follow-up than with baseline factors. Among 20 participants with atrophic gastritis detected at baseline, 19 (95%) had a lower atrophic gastritis severity classification (including none) at follow-up. All 20 of these participants completed treatment to eliminate *Hp* before follow-up and 17 (85%) had improved *Hp* density at follow-up; the small sample size did not permit assessment of factors associated with improved severity of atrophic gastritis.

Of seven participants with intestinal metaplasia detected at baseline, two had a lower intestinal metaplasia severity classification at follow-up, four had the same intestinal metaplasia severity at follow-up and one had a higher severity classification at follow-up. New onsets of intestinal metaplasia were observed in eight participants and new onsets of atrophic gastritis were observed in seven participants; six of these participants had new onsets of both atrophic gastritis and intestinal metaplasia. One participant with no atrophic gastritis and mild intestinal metaplasia at baseline had moderate atrophic gastritis and moderate intestinal metaplasia at follow-up. In total, 10 participants had a higher classification of atrophic gastritis, intestinal metaplasia or both at follow-up. The frequency of progression of atrophic gastritis and/or intestinal metaplasia among all 69 participants assessed at baseline was 14%; this frequency was relatively high in participants who did not complete treatment to eliminate *Hp* before follow-up, did not test negative for *Hp* before follow-up and/or had moderate or marked *Hp* density at follow-up.

Table 5 presents estimated risk differences corresponding to differences in the probability of changed gastric pathology adjusted for confounding factors to the extent possible given the small sample size and high degree of collinearity among variables of interest. Demographic variables, including age, sex, ethnicity, and community showed small differences in probabilities in multivariable models as did time between baseline and follow-up gastroscopy. To select between highly colinear variables, those with stronger independent associations were preferred. In multivariable models of the probability of improved chronic gastritis and improved active gastritis, this probability was greatest among those who had improved *Hp* density at follow-up. For chronic gastritis, the next strongest predictor was completion of treatment to eliminate *Hp* before follow-up; for active gastritis, the next strongest predictor was *Hp* density at baseline, with the probability of improved active gastritis decreasing as baseline *Hp* density increased. For progression of atrophic gastritis and/or intestinal metaplasia, the strongest predictors were detection of *Hp* in gastric biopsies at baseline or follow-up, positively associated with progression, and treatment to eliminate *Hp* before follow-up, inversely associated with progression.

## 4. Discussion

This follow-up study estimated the prevalence of *Hp* infection and associated gastric abnormalities among residents of three Arctic Indigenous communities at time points before and several years after they received treatment to eliminate *Hp* infection. The observed baseline *Hp* prevalence of 72% is in the range of estimates from other northern Indigenous communities in Canada (51–95%), Alaska (80%) and Greenland (58%) [22]. The estimated *Hp* prevalence at follow-up of 26% was substantially lower, in the range of *Hp* prevalence estimates of 20–30% for major urban centers in Canada [4,39].

Similarly, the prevalence of active gastritis, chronic gastritis and atrophic gastritis were all substantially lower at follow-up than baseline. The severity distribution of active, chronic, and atrophic gastritis also showed a notable shift toward reduced severity at follow-up. In particular, there was a substantial reduction in the prevalence of severe chronic gastritis from baseline to follow-up. Examination of within-individual change in gastritis severity shows evidence of the effect of *Hp* infection on gastric pathology. While most participants who were *Hp*-negative at baseline and follow-up had no evidence of abnormal gastric pathology at either time, participants who were *Hp*-positive at baseline and did not receive treatment to eliminate *Hp* showed predominantly unchanged or higher gastric pathology severity at follow-up. Most participants who were *Hp*-positive at baseline and *Hp*-negative at follow-up showed lower severity of gastric pathology at follow-up.

In this study, most participants who had successful treatment to eliminate *Hp* at baseline remained free of infection at follow-up. Among 33 participants who were *Hp*-negative after treatment at baseline, 7 were *Hp*-positive at follow-up, a reinfection proportion of 21% over an average follow-up of 7.2 (median = 9.1) years. For comparison, the estimated reinfection frequency over 2 years following successful *Hp* treatment in a study of urban Alaska Native gastroenterology patients was 14.6% [40]. Authors of the Alaska study report estimated that around a third of the apparent reinfections likely resulted from misclassification of the treatment outcome due to infection that was suppressed to undetectable levels rather than cured, counting those with a first positive UBT within 4 months after the treatment outcome was assessed and two patients with genetic relatedness of *Hp* isolated before and after treatment. Whether reinfections we observed are the result of true reinfection rather than a misclassified treatment outcome cannot be established in our study. In CAN*Help* community projects, however, the estimated prevalence of *Hp* infection in children is similar to that of adults [27], indicating that frequent transmission is ongoing; thus, it is likely that some reinfections occurred.

For active and chronic superficial gastritis and atrophic gastritis, our results are compatible with the hypothesis that *Hp*-associated inflammatory changes of the gastric mucosa are reversible. In contrast, the results for intestinal metaplasia did not follow this pattern, though the small study size should be kept in mind. Most studies that have examined the reversibility of *Hp*-induced histopathological changes following elimination of *Hp* infection show chronic superficial gastritis to be reversible [41,42,43,44,45,46,47,48], but results pertaining to improvement in atrophic gastritis and intestinal metaplasia have shown conflicting results [44,46,47,48,49]. A recent Cochrane review appraising the effectiveness of treatment to eliminate *Hp* for prevention of gastric cancer highlights evidence consistent with the hypothesis that intestinal metaplasia is a “point of no return” in the carcinogenesis pathway [48]. However, one Korean cohort study showed a decrease in the grade of atrophic gastritis at years 1 and 5 of follow-up after *Hp* elimination and a small progressive decrease in the grade of intestinal metaplasia over 5 or more years of annual follow-up after *Hp* elimination [44]. A Colombian chemoprevention trial showed that reversal of atrophic gastritis was common in trial participants who remained free of *Hp* for 6 or more years following baseline therapy; reversal of intestinal metaplasia occurred to a much more limited extent, but *Hp*-negativity was associated with reduced progression of intestinal metaplasia to more advanced disease [50]. In this sense, if there is a “point of no return”, it does not rule out the possibility of gastric cancer prevention in people with intestinal metaplasia.

Notable limitations of this study include the small study size at follow-up, largely due to the discomfort of unanesthetized gastroscopy. Loss-to-follow-up of participants with baseline pathology data was 78% combining all communities. This amount of loss-to-follow-up paired with likely differentiating characteristics of participants who were or were not willing to undergo gastroscopy for a second time may have selected disproportionately for participants who were *Hp*-negative at follow-up with reduced severity of gastric pathology. Also, the alignment of community of residence with length of follow-up prevented us from examining the effect of follow-up duration independent of community. Also, we did not examine diet or smoking as potential confounders because doing so would have further reduced our study size.

We note that some degree of misclassification of histopathology is unavoidable: biopsy sampling can miss *Hp* in the stomach because of its typical patchy colonization; furthermore, *Hp* can be difficult to visualize in tissue slides when bacterial density is low. We also acknowledge that observing *Hp* and gastric pathology status at just two time points could have resulted in missing changes in status that occurred between observations. This is a particular concern for chronic *Hp* infection: given *Hp*’s preferred niche of normal gastric tissue, colonizable tissue disappears as gastric disease progresses, resulting in the disappearance of *Hp* from areas of intestinal metaplasia and more advanced lesions [51]. It is also possible that some participants had undetected *Hp* infection onsets and eliminations during the follow-up period, and a few may have received treatment that was not reported to us.

Despite challenges, there are few population-based studies with long-term follow-up of gastric pathology. Furthermore, we are not aware of any others completed using a community-driven approach. Each project component, from project initiation to data analysis, was guided by local planning committee members who provided local knowledge and insight for grounding the research in the study population’s cultural and physical context; this approach encouraged participation from communities who were otherwise not represented in health research.

## 5. Conclusions

This community-driven research provides encouraging evidence that the elimination of *Hp* infection reduces the severity of *Hp*-associated gastric outcomes over the long term. Such evidence is needed to inform gastric cancer prevention strategies based on screening for *Hp* infection and offering treatment to those who test positive. In general, countries outside of east Asia have not implemented such prevention strategies, despite accumulating evidence that the elimination of *Hp* infection in people at elevated risk of gastric cancer is cost-effective for preventing gastric cancer (along with peptic ulcer disease) [49]. Among assorted reasons for the lack of gastric cancer prevention strategies, there is a lack of information needed to design cost-effective strategies for specific high-risk populations. In this context, the CAN*Help* community-driven projects have generated local evidence of relevance to clinical decision making about *Hp* infection specific to Arctic Indigenous communities. In particular, these projects have shown a high prevalence of severe chronic gastritis and atrophic gastritis in *Hp*-positive participants [27]; high prevalence of the high-risk incomplete cell type among participants with intestinal metaplasia (unpublished data); high rates of treatment success with optimal regimens among participants with good adherence to the regimen [36]; and relatively low reinfection rates [52].

In addition to generating evidence for gastric cancer prevention approaches, we have used the information generated by the CAN*Help* projects to adapt current Canadian guidelines for northern and Indigenous populations in Canada [53]. Our guidelines highlight the ethical imperative that health care practitioners convey to their patients that there is evidence of an increased gastric cancer risk in Arctic and Indigenous populations, along with evidence that curing *Hp* infection reduces gastric cancer risk. We also recommend that endoscopic surveillance be considered for individuals who have intestinal metaplasia or severe atrophy along with another gastric cancer risk factor (Indigenous ethnicity, immigration from a high-incidence region, family history of gastric cancer in a first degree relative particularly if onset age was early, or smoking), especially if the intestinal metaplasia is extensive (in both gastric antrum and body) or severe [53].

When drawing on the community project results to fill in evidence gaps for cancer prevention, due weight should be given to patients’ and communities’ motivation, values, preferences and circumstances, particularly where evidence is limited. Taken in this manner, the value to community members and health care practitioners of what can be learned from a small community-driven follow-up study is evident. Compared to no evidence from the target population, the more frequent observation of improvements in precancerous gastric pathology observed in participants whose *Hp* density reduced during follow-up relative to those whose *Hp* density did not constitutes compelling evidence that treatment to eliminate *Hp* infection has the potential to reduce stomach cancer risk in participating communities.

## Figures and Tables

**Figure 1 cancers-15-03950-f001:**
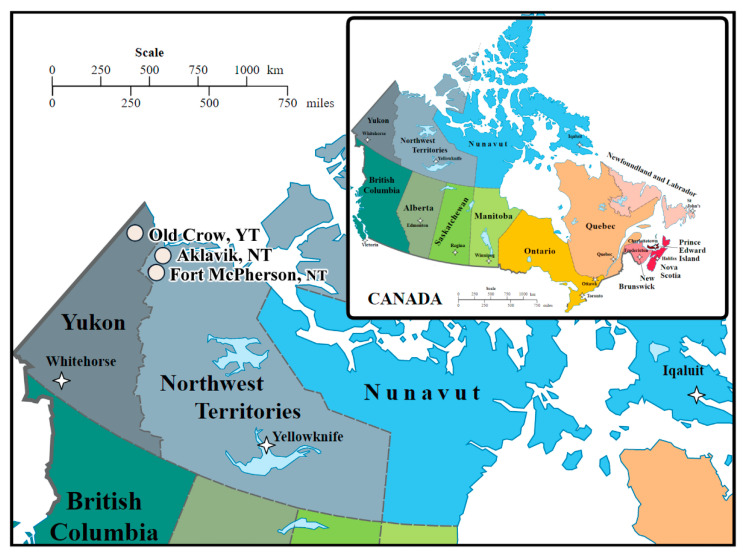
Map of Participating Communities: Aklavik, Northwest Territories (NT); Old Crow, Yukon (YT); and Fort McPherson, Northwest Territories (NT). Adapted from https://commons.wikimedia.org/wiki/File:Political_map_of_Canada.svg (accessed on 8 December 2022).

**Table 1 cancers-15-03950-t001:** Demographics characteristics of gastroscopy participants, at baseline and follow-up.

		Baseline		Follow-Up	
		n	%	n	%
Total		310	100	69	100
Sex					
	Male	139	45	25	36
	Female	171	55	44	64
Ethnicity					
	Non-Indigenous	22	7	3	4
	Inuvialuit	117	40	26	39
	Gwich’in	139	47	36	54
	Other	16	5	2	3
	Missing ***	16		2	
Age at endoscopy (years)					
	<30	76	24	3	4
	30–59	172	56	41	59
	60+	62	20	25	36

* excluded from denominator.

**Table 2 cancers-15-03950-t002:** Prevalence of histopathologic classifications (based on highest grade observed in multiple biopsies) among gastroscopy participants, at baseline and follow-up.

		Baseline (n = 310)			Follow-Up (n = 69)		
		n	%	95% CI *	n	%	95% CI *
*H. pylori* infection		223	72	66, 77	18	26	16, 36
Active gastritis							
	None	93	30	25, 35	51	74	64, 84
	Any	215	70	65, 75	18	26	16, 36
	Mild	101	33	27, 38	12	17	8, 28
	Moderate	81	26	21, 31	6	9	3, 18
	Severe	33	11	7, 14	0	0	0, 5
	Missing **	2			0		
Chronic gastritis							
	None	76	25	19, 29	43	62	51, 74
	Any	234	75	71, 80	26	38	26, 49
	Mild	28	9	6, 12	12	17	8, 26
	Moderate	99	32	27, 37	11	16	7, 25
	Severe	107	34	29, 40	3	4	1, 12
Atrophic gastritis							
	None	214	69	64, 74	59	86	77, 94
	Any	96	31	26, 36	10	14	6, 23
	Mild	67	22	17, 26	8	12	5, 22
	Moderate	24	7	5, 11	2	3	0, 10
	Severe	5	2	0.5, 4	0	0	0, 5
Intestinal metaplasia							
	None	268	86	83, 90	56	81	72, 90
	Any	42	14	10, 17	13	19	10, 28
	Mild	27	9	6, 12	9	13	6, 23
	Moderate	12	4	2, 6	4	6	2, 14
	Severe	3	1	0.2, 3	0	0	0, 5

CI, confidence interval. * wald; binomial exact 95% CI for cell counts < 10 and one-sided 97.5% CI for proportions of 0 or 1. ** excluded from denominator.

**Table 3 cancers-15-03950-t003:** Within-individual change from baseline to follow-up in the severity of gastric pathology, by *Hp* infection status at baseline and follow-up.

		*Hp*− at Follow-Up				*Hp+* at Follow-Up ^†^	
		*Hp*− at Baseline n = 17		*Hp+* at Baseline ^‡^ n = 34		n = 18	
		%	95% CI *	%	95% CI *	%	95% CI *
Chronic gastritis							
	None at both times	65	42, 87	0	0, 10	0	0, 19
	Lower	12	1, 36	100	90, 100	39	17, 64
	Unchanged	6	0.1, 29	0	0, 10	44	22, 69
	Higher	18	4, 43	0	0, 10	17	4, 41
Active gastritis							
	None at both times	100	80, 100	3	0.1, 15	0	0, 19
	Lower	0	0, 20	97	91, 100	28	10, 53
	Unchanged	0	0, 20	0	0, 10	50	26, 74
	Higher	0	0, 20	0	0, 10	22	6, 48
Atrophic gastritis							
	None at both times	88	73, 100	59	42, 75	39	17, 64
	Lower	0	0, 20	38	22, 56	33	13, 59
	Unchanged	0	0, 20	0	0, 10	6	0.1, 27
	Higher	12	1, 36	3	0.1, 15	22	6, 48
Intestinal Metaplasia							
	None at both times	88	73, 100	88	77, 99	50	26, 74
	Lower	0	0, 20	3	0.1, 15	6	0.1, 27
	Unchanged	0	0, 20	3	0.1, 15	17	4, 41
	Higher	12	1, 36	6	1, 20	28	10, 53

* wald; binomial exact 95% CI for cell counts < 10 and one-sided 97.5% CI for proportions of 0 or 1; distributions do not all sum 100% due to rounding to the nearest even number; ^†^ includes seven participants with successful treatment at baseline, seven with failed treatment at baseline, three with no treatment at baseline and one classified as negative at baseline; ^‡^ includes 26 participants with successful treatment at baseline, 6 with failed treatment at baseline and 2 with no treatment at baseline.

**Table 4 cancers-15-03950-t004:** Probability (risk) of changes in gastric pathology by selected factors.

		n	Risk	95% CI *
Chronic gastritis improved *among 54 participants with chronic gastritis at baseline*				
Total		43/54	0.80	0.66, 0.89
Sex				
	Male	16/21	0.76	0.53, 0.92
	Female	27/33	0.82	0.65, 0.93
Age (years)				
	27–49	14/19	0.74	0.49, 0.91
	50–59	14/15	0.93	0.68, 1.0
	60–78	15/20	0.75	0.51, 0.91
Ethnicity				
	Gwich’in	26/30	0.87	0.69, 0.96
	Inuvialuit	14/19	0.74	0.49, 0.91
Community				
	Aklavik NT	21/27	0.78	0.58, 0.91
	Old Crow YT	8/9	0.89	0.52, 1.00
	Fort McPherson NT	14/18	0.78	0.52, 0.94
Baseline *Hp* density				
	No bacteria observed	2/3	0.67	0.09, 0.99
	Mild	12/17	0.71	0.44, 0.90
	Moderate	18/21	0.85	0.64, 0.97
	Marked	11/13	0.85	0.55, 0.98
Baseline *Hp* resistance				
	To metronidazole	13/17	0.76	0.49, 0.92
	To clarithrymycin	7/7	1	0.53, 1
	To 1+ antibiotics	15/19	0.79	0.53, 0.92
	To 2+ antibiotics	5/6	0.83	0.23, 0.99
Baseline chronic gastritis severity				
	Mild	4/8	0.50	0.16, 0.84
	Moderate	18/23	0.78	0.56, 0.93
	Severe	21/23	0.91	0.72, 0.99
Treated to eliminate *Hp* before follow-up				
	No	3/6	0.50	0.12, 0.88
	Yes	40/48	0.83	0.70, 0.93
*Hp*-negative breath test before follow-up				
	No	12/18	0.67	0.41, 0.87
	Yes	31/36	0.86	0.71, 0.95
*Hp* density improved at follow-up				
	No	4/13	0.20	0.03, 0.56
	Yes	39/41	0.95	0.84, 0.99
*Hp* density at follow-up				
	No bacteria observed	36/37	0.97	0.86, 1.0
	Mild	2/5	0.40	0.05, 0.85
	Moderate	4/9	0.44	0.14, 0.79
	Marked	0/2	0	0, 0.85 ^†^
Active gastritis improved *among 50 participants with active gastritis at baseline*				
Total		36/50	0.72	0.56, 0.84
Sex				
	Male	14/20	0.70	0.46, 0.88
	Female	22/30	0.73	0.54, 0.88
Age (years)				
	27–49	10/19	0.53	0.29, 0.76
	50–59	13/14	0.93	0.66, 1.00
	60–78	13/17	0.76	0.50, 0.93
Ethnicity				
	Gwichin	21/27	0.78	0.58, 0.91
	Inuvialuit	14/18	0.78	0.52, 0.94
Community				
	Aklavik NT	18/26	0.69	0.48, 0.86
	Old Crow YT	6/9	0.67	0.30, 0.93
	Fort McPherson NT	12/15	0.80	0.52, 0.96
Baseline *Hp* density				
	Mild	12/16	0.75	0.48, 0.93
	Moderate	16/21	0.76	0.53, 0.92
	Marked	8/13	0.62	0.32, 0.86
Baseline *Hp* resistance				
	To metronidazole	11/17	0.65	0.38, 0.84
	To clarithrymycin	7/7	1	0.53, 1
	To 1+ antibiotics	13/19	0.68	0.43, 0.86
	To 2+ antibiotics	6/6	1	0.48, 1
Baseline active gastritis severity				
	Mild	20/27	0.74	0.54, 0.89
	Moderate	12/18	0.67	0.41, 0.87
	Marked	4/5	0.80	0.28, 0.99
Treated to eliminate *Hp* before follow-up				
	No	3/4	0.75	0.19, 0.99
	Yes	33/46	0.72	0.57, 0.84
*Hp*-negative breath test before follow-up				
	No	10/18	0.56	0.31, 0.78
	Yes	26/32	0.81	0.64, 0.93
*Hp* density improved at follow-up				
	No	3/10	0.30	0.07, 0.65
	Yes	33/40	0.83	0.67, 0.93
*Hp* density at follow-up				
	No bacteria observed	33/33	1	0.89, 1 ^†^
	Mild	0/5	0	0, 0.52 ^†^
	Moderate	7/9	0.22	0.03, 0.60
	Marked	2/4	0.50	0.01, 0,99
Atrophic gastritis improved *among 20 participants with atrophic gastritis at baseline*				
Total		19/20	0.95	0.75, 1 ^†^
Treated to eliminate *Hp* before follow-up				
	No	0/20	--	--
	Yes	20/20	0.95	0.75, 1 ^†^
*Hp*-negative breath test before follow-up				
	No	6/6	1	0.54, 1 ^†^
	Yes	13/14	0.93	0.66, 1.00
*Hp* density improved at follow-up				
	No	3/3	1	0.29, 1 ^†^
	Yes	16/17	0.94	0.71, 1.00
Atrophic gastritis and/or intestinal metaplasia progressed (including new onsets and six co-occurring cases) *among 69 participants assessed at baseline*				
Total		10/69	0.14	0.07, 0.25
Sex				
	Male	3/25	0.12	0.03, 0.31
	Female	7/44	0.16	0.07, 0.30
Age (years)				
	27–49	5/21	0.24	0.08, 0.47
	50–59	1/23	0.04	0.00, 0.22
	60–78	4/25	0.16	0.05, 0.36
Community				
	Aklavik NT	9/39	0.23	0.11, 0.39
	Old Crow YT	0/9	0	0, 0.34 ^†^
	Fort McPherson NT	1/21	0.05	0.00, 0.24
Baseline *Hp* density				
	No bacteria observed	3/18	0.17	0.04, 0.41
	Mild	2/17	0.12	0.01, 0.36
	Moderate	4/21	0.19	0.05, 0.42
	Marked	1/13	0.08	0.00, 0.36 ^†^
Treated to eliminate *Hp* before follow-up				
	No	5/21	0.24	0.08, 0.47
	Yes	5/48	0.10	0.03, 0.23
*Hp*-negative breath test before follow-up				
	No	4/18	0.22	0.06, 0.48
	Yes	6/51	0.12	0.04, 0.24
*Hp* density at follow-up				
	No bacteria observed	4/51	0.08	0.02, 0.19
	Mild	0/6	0	0, 0.46 ^†^
	Moderate	5/10	0.50	0.19, 0.81
	Marked	1/2	0.50	0.01, 0.99

CI, confidence interval; NT, Northwest Territories; YT, Yukon; *Hp*, *H. pylori*; * exact binomial confidence interval; ^†^ one-sided, 97.5% exact binomial confidence interval.

**Table 5 cancers-15-03950-t005:** Differences in the probability (risk differences) of changed gastric pathology estimated using multivariable binomial regression.

				* Adjusted Risk Difference	95% CI
Probability of improved chronic gastritis *among 54 participants with chronic gastritis at baseline*					
Treated to eliminate *Hp* at baseline					
	No			Referent	
	Yes			0.29	0.04, 0.53
*Hp* density improved at follow-up					
	No			Referent	
	Yes			0.74	0.50, 0.99
Probability of improved active gastritis *among 50 participants with active gastritis at baseline*					
Baseline *Hp* density					
	Mild			Referent	
	Moderate			−0.15	−0.31, −0.01
	Marked			−0.35	−0.58, −0.13
*Hp* density improved at follow-up					
	No			Referent	
	Yes			0.63	0.39, 0.87
Progression in atrophic gastritis and/or intestinal metaplasia (including new onsets and six co-occurring cases) *among 69 participants assessed at baseline*					
*Hp* observed on histology at baseline					
	No			Referent	
	Yes			0.19	0.12, 0.27
Treated to eliminate *Hp* at baseline					
	No			Referent	
	Yes			−0.29	−0.45, −0.13
*Hp* observed on histology at follow-up					
	No			Referent	
	Yes			0.23	0.00, 0.45

* Adjusted for variables listed in table under the corresponding outcome subheading.

## Data Availability

The datasets used for the study are available from the CAN*Help* Working Group (canhelp@ualberta.ca) on reasonable request following community review of proposed data uses.

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
