# Peer review of "Changes in Gastric Pathology after H. pylori Treatment in Community-Driven Research Aimed at Gastric Cancer Prevention"

_cancers, 2023, doi:10.3390/cancers15153950_

Round 1

Reviewer 1 Report (Previous Reviewer 2)

I absolutely do not agree with the comment to point 3. p values MUST be added to tables. All other answers were fine.

Author Response

Reviewer 2 Report (New Reviewer)

The article by Ting Wang et al. shows the histopathological differences after elimination of H. pylori in very specific cancer populations. 

With no novelty, apart from the geographical location, this article could be published after revision if the editor deems it relevant.

Passive language is preferred.

Numbers less than or equal to 12 should be written out in full.

The method section should be divided into sub-sections to aid comprehension/reading.

The absence of an antibiogram is very detrimental to the results of the study.

Why is the test used one-sided?

More than the change at the time of the control compared with the initial event, the change as a function of the time between the two samples should be analysed.

How did the authors take into account the risk of inflation of the alpha risk, given the multiplicity of statistical tests used?

Author Response

This manuscript is a resubmission of an earlier submission. The following is a list of the peer review reports and author responses from that submission.

Round 1

Reviewer 1 Report

The presented manuscript has a very interesting theme, but it does not explore it in an objective way and with the methodological rigor that is expected for publication in Cancers.

The importance of the present study in the selected population was not explored. No other potential risk factors for gastric cancer were identified in the participating communities, for example, the type of diet.

The authors group the baseline individuals from the three locations in the analyses, but they seem to be very different studies, especially the one carried out in 2008 in Aklavik. The Follow-up time period is twice as long as the others. As the authors do not separate the three participating communities, perhaps there is an important dilution of the possible findings. Another point of methodological laxity is the union of Hp+ “successfully treated” individuals, with “treatment failures” and “not treated” for comparative analysis between groups in the tables.

The conclusion is extensive and tiring, in addition to including conclusions that go beyond the manuscript presented here.

There is a lack of information about the collections, the reagents used and the methodologies of the diagnostic tests. Once Hp is diagnosed, it is not considered or categorized according to the type and success of the treatment or the virulence of the bacteria.

Finally, the methodology used does not allow access to the objectives and does not match the title of the manuscript.

Reviewer 2 Report

In the present original article Wang et al showed that in native populations of Canadian Arctic communities, a program of H. pylori diagnosis and treatment and upper endoscopy follow up reduced the severity of gastritis and the incidence of precancerous lesions. Main comments:

1) Considering the low-density population of enrolled regions, which population proportion is represented by included patients?

2) At follow up gastroscopy, 18 patients were H. pylori positive. Did they fail eradication at baseline?

3) Please add p values in Table 2.

4) Confidence interval should be separated by a -dash , not by a comma.

5) A Methods paragraph describing statistical analysis is lacking.

6) It seems that no dysplastic lesions were observed, is it correct?

7) How can the Authors justify the high prevalence of precancerous lesions in natives? May some ambiantal factors (fish smoking cooking) contribute?